# Influence of the Crystal Forms of Calcium Carbonate on the Preparation and Characteristics of Indigo Carmine-Calcium Carbonate Lake

**DOI:** 10.3390/foods13162607

**Published:** 2024-08-20

**Authors:** Le Jing, Yuhan Liu, Jiaqi Cui, Jinghan Ma, Dongdong Yuan, Chengtao Wang

**Affiliations:** Beijing Engineering and Technology Research Center of Food Additives, School of Food and Health, Beijing Technology & Business University, Beijing 100048, China; jingle1998@163.com (L.J.); liuyuhan_210@163.com (Y.L.); c18647362147@163.com (J.C.); jjh02522@126.com (J.M.); wctbtbu@hotmail.com (C.W.)

**Keywords:** calcium carbonate, indigo carmine, polymorphism, colorant lake, adsorption

## Abstract

In this study, indigo carmine (IC)-calcium carbonate lakes with different crystalline forms of calcium carbonate were prepared through co-precipitation methods, and the properties of these lakes and their formation mechanisms were investigated. The results showed that amorphous calcium carbonate (ACC) exhibited the smallest particle size and the largest specific surface area, resulting in the highest adsorption efficiency. Vaterite, calcite, and aragonite followed after ACC in decreasing order of adsorption efficiency. Kinetic analysis and isothermal analysis revealed the occurrence of chemisorption and multilayer adsorption during formation of the lakes. The FTIR and Raman spectra suggested participation of sulfonic acid groups in chemisorption. Appearance of IC significantly altered TGA curves by changing weight loss rate before decomposition of calcium carbonate. EDS analysis revealed the adsorption of IC predominantly happened on the surface of calcium carbonate particles rather than the interior.

## 1. Introduction

Indigo, one of the oldest known pigments [1], has limited solubility in water. To address this limitation, indigo carmine (IC), a sulfonic acid derivative of indigo, has found widespread use in industries such as food, medicine, printing, and dyeing [2]. IC, also recognized as Food Blue No.1, Food Cyan No.2, or simply Food Blue, is notable for its high solubility in water. It retains the characteristic blue color of indigo, rendering it a stable non-azo colorant [3,4,5]. Colorant lakes containing water-soluble colorants like IC enhance stability and dyeing. Traditionally, aluminum hydroxide is the substrate for providing food-grade colorant lakes [6]. However, aluminum has been linked to diseases such as osteochondrosis and neurological disorders in humans. When aluminum is injected directly into the brains of animals or accidentally enters the human brain through, for example, dialysis, it can be neurotoxic, leading to the neurological syndromes of dialysis encephalopathy or dialysis dementia. Cognitive and other neurological deficits may exist if groups are occupationally exposed to high concentrations of aluminum dust [7,8]. Reducing aluminum intake from food sources has become a recent focus in the food industry and academia. Our team suggests calcium carbonate-based colorant lakes as substitutes of aluminum hydroxide ones [9].

Calcium carbonate is a common food additive with a wide range of application for supplements, coloring, bulking, and antacids [10,11,12]. Calcium carbonate solid has one amorphous form and three crystal forms, including calcite, aragonite, and vaterite. Amorphous calcium carbonate (ACC) is the least stable form of calcium carbonate but has the highest surface area, which renders its high adsorption capacity. Synthetic calcium carbonate mixture often yields a more stable crystalline form. Our team has successfully developed a *Monascus* pigments (MPs)-calcium carbonate lake. This lake demonstrates significantly enhanced light stability compared to pure MPs [13].

In this study, the initial preparation of calcium carbonate colorant lakes with varied crystalline forms was accomplished through co-precipitation, marking a significant advancement. The fundamental properties of ACC, the three polymorphs, and their colorant lakes, including zeta potential, particle size, BET-specific surface area, and color stability were investigated. Then, scanning electron microscope (SEM) and X-ray diffraction (XRD) techniques were used to analyze the micro-morphological characteristics of calcium carbonate and lakes. Next, the formation of IC-calcium carbonate complexes was investigated through kinetic and isothermal adsorption analysis. Moreover, the state of IC and calcium carbonate within the lakes and their interaction were investigated using Fourier-infrared spectroscopy (FTIR), Raman spectroscopy, thermogravimetric analysis and differential scanning calorimetry (TGA-DSC) technique. Finally, energy-dispersive X-ray spectroscopy (EDS) was utilized to determine the distribution of IC within the lake. This research shed light on the formation mechanism of calcium carbonate-based colorant lakes and suggests ways to enhance their quality.

## 2. Materials and Methods

### 2.1. Chemicals

IC (96% *w*/*w*) was purchased from Macklin Reagent Co., Ltd. (Shanghai, China) and used as received. All other chemicals, such as calcium chloride, sodium carbonate, ethanol, sodium hydroxide, and hydrochloric acid, were of analytical grade and obtained from local suppliers in China. The deionized (DI) water (~18.25 MΩ·cm) used in solution preparation in this study was obtained from the laboratory’s water purification system (HYP-QX-UP, Huiyipu Ltd., Beijing, China).

### 2.2. Preparation of Calcium Carbonate with Different Crystalline Forms

The methods employed for the preparation of calcium carbonate were adapted from Nebel & Epple [14], Trushina, Bukreeva, & Antipina [15], and Zou et al. [16]. However, these methods were systematically modified in a preliminary investigation with the objective of optimizing the purity of each calcium carbonate.

Calcite: A mixture of 150 mL of 0.2 mol/L CaCl_2_ and 150 mL of 0.2 mol/L Na_2_CO_3_ was stirred at 600 r/min at ambient temperature for 30 min. The resulting solution was centrifuged (5000 r/min for 5 min) to obtain sediment. The sediment was then dried in an oven at 45 °C for 12 h and ground to obtain IC—calcite lake.Aragonite: 150 mL of 0.2 mol/L CaCl_2_ and 150 mL of 0.2 mol/L Na_2_CO_3_ were preheated to 80 °C in a water bath, respectively. After rapid mixing, the solution was centrifuged (5000 r/min for 5 min) to obtain sediment. The sediment was dried in an oven at 45 °C for 12 h and ground to obtain IC—aragonite lake.Vaterite: A mixture of 150 mL of 0.1 mol/L CaCl_2_ (60% DI water + 40% ethanol) and 150 mL of 0.1 mol/L Na_2_CO_3_ solution (60% DI water + 40% ethanol) was stirred at 600 r/min at ambient temperature for 30 min. After centrifuging the solution at 5000 r/min for 5 min to obtain sediment, the sediment was dried in an oven at 45 °C for 12 h and ground to obtain IC—vaterite lake.ACC: A rapid mixture of 20 mL of 0.1 mol/L CaCl_2_ solution, 20 mL of 0.1 mol/L Na_2_CO_3_ solution, and 40 mL of ethanol was introduced into the filter cup of a vacuum membrane filtration set. After agitating the filter cup until many particles appeared in the reaction solution, a 250 mL of ethanol was rapidly added and vacuum-filtered. The resulting solid residue was dried in a vacuum oven at 45 °C for 12 h, and ground to obtain IC—ACC lake.

### 2.3. Preparation of Calcium Carbonate Colorant Lakes

IC was dissolved in sodium carbonate, and calcium carbonate was synthesized as in Section 2.2 to make lakes. Three colorant lakes with different IC additions were prepared for each of the four calcium carbonates. IC was added accordingly at CaCO_3_: IC ratios of 3 g:50 mg, 3 g:200 mg, and 3 g:500 mg, respectively. Specific formulations of samples are provided in Appendix A.

### 2.4. Characterization of CaCO_3_ and Its Colorant Lakes

SEM and EDS: the morphological characteristics of calcium carbonate and colorant lakes were examined using a scanning electron microscope (Hitachi S-4800, Hitachi, Ltd., Tokyo, Japan). For surface observation, the samples were directly sputtered with gold before observation. For interior observation, powdered samples were frozen in liquid nitrogen, ground with a mortar and pestle, and sputtered with gold. Elemental distribution on the surface and interior of the particles was analyzed using spot-scanning mode in EDS analysis.

XRD: qualitative and quantitative determinations of the crystal composition in calcium carbonate and the colorant lakes were conducted. Calcium carbonate and colorant lakes were scanned using an X-ray diffractometer (Rigaku SmartLab SE, Tokyo, Japan) in fine scanning mode, with scanning speed of 2°/min and a scanning range of 2θ from 5–90°. Data analysis was conducted using MDI Jade 6.5.

Zeta potential and particle size: surface charge of particles in samples were analyzed using a zeta potential meter. The prepared samples were dispersed in DI water, and 1 mL of the suspension was injected into a capillary zeta potential cell and measured using a Zetasizer Nano ZS90 (Malvern Instruments, Malvern, UK). The particle size of the samples was determined using a Mastersizer 2000 (Malvern Instruments, UK), with refractive indices of 1.656 for calcium carbonate and 1.33 for water. All samples underwent testing in triplicate, and the volume-weighted mean diameter, d_4,3_, was recorded.

BET test (multi-point Brunauer-Emmett-Teller specific surface area test): the tests were performed using an Autosorb iQ (Quantachrome, Houston, TX, USA). The samples were adsorbed with nitrogen and subsequently degassed at a 120 °C for 6 h to remove any residual gases. The specific surface area of the samples was then determined.

Light stability: the blue color value of the samples was assessed using a colorimeter (Konica Minolta CM-3610A, Tokyo, Japan). The powder of each sample was put into a small ziplock plastic bag, spread, and gently pressed to create a flat surface for testing. The b values of the samples were measured before and after 24 h of light incubation (12,000 LX) at 45 °C (BSG-300, Boxun, Shanghai, China). The change in the b* value (Δb*) was determined by subtracting the post-illumination b* value from the pre-illumination b* value. IC powders, along with samples created by directly blending IC with each form of calcium carbonate at a ratio of 3 g CaCO_3_ to 500 mg IC, also underwent light stability testing for comparative analysis.

### 2.5. Kinetic Analysis of Adsorption

Co-precipitation mode: kinetic experiments were conducted for the four kinds of calcium carbonate. For each type, CaCl_2_ solution and Na_2_CO_3_ solution were prepared following the procedure outlined in Section 2.2. The timing started simultaneously when IC (CaCO_3_:IC = 2 g:50 mg) was dissolved in sodium carbonate solution and mixed with CaCl_2_ solution (and ethanol for vaterite). The final volume after mixing was 200 mL. The reaction solutions were continuously stirred using a magnetic stirrer (600 r/min) and sampled sequentially with a disposable syringe for about 1.5 mL at predetermined time internals. Then, the samples were filtered using a 0.22 μm filter membrane. The concentration of IC in the filtrate obtained at each time point, labeled C_t_ (mg/L), was measured at 610 nm using UV-Vis spectrometry method [17], and the amount of IC adsorbed by calcium carbonate was calculated at each time point. Then, the amount of IC adsorbed per unit mass of calcium carbonate, labeled q_t_ (mg/g), was calculated using Equation (1), and the curve of q_t_ versus time was plotted. Equations (2) and (3) show pseudo-first- and pseudo-second-order models for the kinetic curves. Finally, the adsorption ratio at each time point was calculated using Equation (4):(1)$$q_{t}=\frac{{IC}_{T}-C_{t}V}{M_{{CaCO}_{3}}}$$
(2)$$\ln\left(q_{e}-q_{t}\right)=lnq_{e}-k_{1}t$$
(3)$$\frac{t}{q_{t}}=\frac{t}{q_{e}}+\frac{1}{k_{2}q_{e}^{2}}$$
(4)$$Adsorption\;ratio=\frac{{IC}_{T}-C_{t}V}{{IC}_{T}}$$
where IC_T_ (mg) is the total mass of IC added in the experiment, V (L) is the volume of the reaction solution, $M_{{CaCO}_{3}}$ (g) is the mass of CaCO_3_, q_e_ (mg) is the amount of the adsorbate adsorbed per unit mass of adsorbent at equilibrium, q_t_ (mg) is the amount of the adsorbate adsorbed per unit mass of adsorbent at time t, k_1_ (min^−1^) is the pseudo-first-order rate constant and k_2_ (g/(mg·min)) is the pseudo-second-order rate constant.

Surface adsorption mode: a specific quantity of each of the four calcium carbonate types prepared in Section 2.2 was weighted and dispersed in the IC solution (calcium carbonate was added at a ratio of CaCO_3_:IC = 2 g:50 mg) to achieve surface adsorption. The total volume after mixing was adjusted to 200 mL. The subsequent procedures, encompassing both the collection of samples and the kinetic analysis, were carried out identically to those employed in the co-precipitation method.

### 2.6. Isothermal Analysis of Adsorption

The two reaction solutions for preparing each of the four forms of calcium carbonates were prepared according to the method described in Section 2.2. Subsequently, the two solutions were rapidly mixed into a 250 mL triangular flask and subjected to magnetic stirring at 600 r/min, with the timing initiated simultaneously. Samples were taken and filtered through a 0.22 μm membrane every 30 min. After which, more IC was added to the reaction system immediately. The concentration of IC in the filtrate, C_e_ (mg/L), was measured using the method described in Section 2.5. The experiment was ended when the reaction solution was saturated with IC. The amount of IC adsorbed by calcium carbonate at each time point was calculated using Equation (5) to find the mass of IC adsorbed per unit mass of calcium carbonate at equilibrium, labeled q_e_ (mg/g), and the curve of q_e_ versus C_e_ was plotted. The curves were fitted into the Freundlich model (Equation (6)) and the Langmuir model (Equation (7)).

Due to rapid crystallization of ACC in aqueous solutions, the isothermal experiment for ACC cannot be initiated with only one calcium carbonate preparation at beginning of the test. Thus, for each addition of IC, a new ACC was prepared using the method described in Section 2.2, and the q_e_ and C_e_ were determined at each solution for plotting.
(5)$$q_{e}=\frac{{IC}_{T}-C_{e}V}{M_{{CaCO}_{3}}}$$
(6)$$lnq_{e}=lnK_{f}+\frac{1}{n}lnC_{e}$$
(7)$$\frac{C_{e}}{q_{e}}=\frac{1}{q_{m}k_{l}}+\frac{C_{e}}{q_{m}}$$
where IC_T_, V and $M_{{\mathrm{CaCO}}_{3}}$ are the same as defined above, K_f_ (mg^1−1/n^·g^−1^·L^1/n^) is the Freundlich constant characterizing a particular adsorption isotherm, n (dimensionless) is Freundlich constant representing adsorption intensity, q_m_ (mg/g) is the q_e_ for a complete monolayer and k_l_ (L/mg) is sorption equilibrium constant.

### 2.7. FTIR Analysis

The FTIR technique was used to analyze the possible bonding force between IC and calcium carbonate in the lakes. Before testing, the samples were dried overnight in a 60 °C oven for 12 h. Pellets of IC, calcium carbonate and lakes were prepared by pressing them with potassium bromide, and these pellets were then subjected to analysis using a Nicolet IS5 spectrometer (Thermo Fisher, Waltham, MA, USA) with a wavenumber range of 400–4000 cm and a resolution of 2 cm^−1^.

### 2.8. Raman Spectra Analysis

Raman spectra of IC, calcium carbonate, and lakes were obtained using a Raman spectrometer (HORIBA LabRAM, Odyssey, Job, France). The excitation wavelength employed was 532 nm, covering a test range of 0–3000 cm^−1^.

### 2.9. TGA-DSC Analysis

The thermal properties of calcium carbonate and colorant lakes were analyzed using a TGA-DSC instrument (TA SDT Q600, New Castle, DE, USA) within the range of 30–800 °C. The powdered samples were placed into an alumina crucible, which was then purged with nitrogen (N_2_). A consistent heating rate of 10 °C per minute was upheld throughout the procedure.

### 2.10. Statistical Analysis

The data were processed and graphed using Origin 2018 (v.9.5). All assays were conducted in triplicate, and the results were expressed as mean ± standard deviation (SD). Means were compared through one-way ANOVA test using IBM SPSS Statistics 26, while differences were considered significant when *p* < 0.05.

## 3. Results and Discussion

### 3.1. Characterization of CaCO_3_ and Colorant Lakes

Figure 1 shows SEM images of the four forms of calcium carbonate and their respective lakes, with the four kinds of calcium carbonate in the first column. Calcite displays typical rhombohedral structures with cracked surfaces, resembling stacked shales, consistent with findings reported by Rodriguez-Blanco et al. [18]. Aragonite is rod-shaped with varying lengths, and its surface resembles dry and cracked hair. Vaterite particles resemble grape beads, forming clusters with appearance of large cavities, as also reported by Nebel et al. [14]. Compared to other forms, ACC exhibits much smaller spherical particles, aggregating to form a relatively compact structure consistent with the literature reports [19,20]. The addition of IC to the forming solution of each calcium carbonate did not noticeably change their micro-morphology, except for the introduction of bar structures (in red circle) into calcite, vaterite, and ACC. As confirmed by Appendix A, the bar structure is a typical feature of IC particle, indicating that a few IC particles could form when high concentration of IC is used to fabricate the lakes.

XRD was used to analyze the crystalline composition of calcium carbonate and colorant lakes. The XRD diffractogram of IC is shown in Appendix A, and it is confirmed that characteristic peaks of IC crystal are mainly scattered in the 2θ range from 5 to 30. Figure 2 shows that in the case of calcite, the peaks of calcite remain unchanged after adding IC to form a calcite lake, while characteristic peaks of IC emerge in the diffractogram of calcite lake. The results confirm the high purity (100%) of calcite by showing no characteristic peak of any other crystal form in its diffractogram. The diffractogram of calcite lake indicates that a crystal solid of IC formed in the lake, consistent with the bar structure marked by red circles in the SEM figure of calcite (Figure 1). Regarding aragonite, apart from its characteristic peaks, a peak belonging to calcite also appeared. In the diffractogram of aragonite lake, one peak belongs to calcite, and three peaks belongs to vaterite, indicating slight interference by IC in the crystallization of aragonite. However, no characteristic peak of IC is present in the diffractogram of aragonite lake. In the diffractogram of vaterite, a few characteristic peaks belonging to calcite appear alongside vaterite peaks, suggesting the presence of detectable amounts of calcite formed during vaterite formation. However, quantitative analysis showed that vaterite still accounted for more than 92% of the composition, still showing high purity. For ACC and ACC lake, no crystal peak is observed in their diffractogram, consistent with expectations. Similar to calcite and vaterite, characteristic peaks of IC crystal solid also appear in the diffractogram of ACC lake. The results were consistent with the observations from SEM as shown in Figure 1. The characteristic peaks of IC appeared in all colorant lakes except aragonite lake. As shown in Section 2.2, only aragonite was prepared at a high temperature (80 °C), suggesting that high temperature could be the key factor inhibiting crystal solid formation of IC.

Calcium carbonate crystals, like all crystals, contain lattice defects. When a lattice ion is vacant or replaced by a foreign ion, the electrical neutrality is broken, generating residual charges [21]. The zeta potential of all four forms of calcium carbonate was negative (Table 1). Calcite exhibited the lowest zeta potential, while ACC displayed the highest zeta potential, approaching zero, which implies that crystal calcium carbonates are more easily able to accumulate negative charges. Interestingly, the potentials of calcite, aragonite, and ACC were lower than those of their respective colorant lakes, while the potential of vaterite was higher than that of the vaterite lake. Positively charged groups (-Ca^+^, -Ca (OH_2_)^+^) and negatively charged groups (-CO_3_^−^, -CO_3_ (OH_2_)^−^) exist on the surface of calcium carbonate, with zeta potential depending on the relative amounts of these positive and negative charges [13]. It is reasonable to expect that the variations in lattice structures among different calcium carbonates would result in distinct charged groups on their surfaces, subsequently causing variation in zeta potential. Theoretically, added IC would bond its sulfonic acid groups (negatively charged) with positive groups on lake’s surface or interior, reducing zeta particle potential. However, only vaterite and its lake were consistent with this theory, exhibiting lower zeta potential in vaterite lake. This phenomenon suggests that upon adding IC, the IC may influence the lattice structure of calcium carbonate, leading to the formation of more positively charged groups rather than solely forming electrostatic bonding between them.

ACC is the precursor of calcium carbonate crystal phases, exhibiting an average particle size of approximately 368.3 nm (Table 1), which is close to the experimental results from Tobler et al. [22]. The particle diameter of the vaterite lake exhibited a reduction with the incremental addition of IC, aligning with prior research where *Monascus* pigments similarly reduced the size of calcium carbonate particles [13]. While the trend was not as evident in the aragonite and calcite lakes, a comparison between the pure calcium carbonates and their respective lakes revealed notable disparities in particle diameter. This suggests that the addition of IC has a significant impact on the particle size across all examined crystal forms.

BET analysis revealed that ACC possessed a significantly larger specific surface area than the crystalline forms, laying the foundation for its superior adsorption properties [23,24]. Vaterite exhibited the largest specific surface area among the crystalline forms due to its grape-bead-like structure.

To evaluate the stability of the blue color in colorant lakes, Δb* values of the colorant lakes before and after 24 h of light illumination were compared as shown in Table 1. Smaller absolute Δb* value indicates improved stabilization. The absolute value of Δb* for IC was greater than that of all the colorant lakes. This indicates that the colorant lakes have higher light stability than the pure IC, verifying the significance of using lake as a substitute for pigments. There was no significant difference in the Δb* values of aragonite lake, neither in that of ACC lakes. However, significant differences are identified among the Δb* values of vaterite lakes and calcite lakes with varying IC contents. The results indicate that the variation of IC contents does not significantly alter the light stability of aragonite lake and ACC lake, but does affect the light stability of vaterite lake and calcite lake. Among all the tested samples, the calcite lake with the lowest IC content exhibited the smallest absolute Δb* value, confirming highest light stability. Control samples were prepared by directly mixing each form of calcium carbonate with IC at ratio of 500 mg IC per 3 g CaCO_3_, which was the same ratio used in preparing calcite lake (500 mg), aragonite lake (500 mg), vaterite lake (250 mg) and ACC lake (33.3 mg). The Aragonite +IC sample exhibited significantly lower Δb* value compared to that of aragonite lake, Vaterite + IC sample showed no significant difference in absolute Δb* value compared to vaterite lake, and Calcite + IC and ACC + IC samples showed significantly higher absolute Δb* value compared to that of their corresponding lakes. In conclusion, the stability of the colorant lakes is superior to that of pure IC, which provides a robust theoretical foundation for the utilization of colorant lakes.

### 3.2. Kinetic Adsorption Analysis

During the process of precipitating calcium carbonate from calcium chloride and sodium carbonate, calcium carbonate nanoparticles firstly form in the reaction solution. Subsequently these nanoparticles aggregate to form micrometer-sized amorphous particles, which then begin to precipitate. Eventually, these amorphous particles undergo spontaneous transformation into a crystalline structure [25]. The kinetic adsorption curve of IC on calcite (Figure 3a) displays a maximum q_t_ value of 10.9 mg/g at 0 min. In the initial stage of the reaction, the curve exhibits a gradual downward trend, with fluctuations around 7 mg/g observed after 15 min. At the very beginning, calcium carbonate was in an amorphous state, thus exposing most binding sites, which resulted in the highest q_t_ values. However, as the unstable amorphous state transformed into calcite, it released the adsorbed IC molecules back into the reaction solution, leading to a decrease in q_t_ value. In fact, adsorption and desorption occurred simultaneously and reach equilibrium after a certain time. The kinetic curves of vaterite (Figure 3c) and ACC (Figure 3d) closely resembles that of calcite, showing a maximum q_t_ at beginning of the curve followed by a gradual decrease until the end of the curve. This is shown more visually on the graph of adsorption ratios (Figure 3f). While calcite and vaterite had similar q_t_ values of approximately 6 mg/g at the end of the test, ACC displayed a final q_t_ of about 2.5 mg/g. Before crystallization, calcium carbonate existed in the form of ACC, which served as the precursor phase of crystalline forms. ACC’s particles, being small and having a large specific surface area compared to the three crystal forms (Table 1), theoretically exposes more adsorption sites on their surface. However, ACC exhibited the lowest adsorption capacity in the test, showing the lowest final q_t._ This unexpected result could be ascribed to the utilization of much lower IC concentration in the reaction solution of ACC lake. As described in Section 2.3, a low concentration of IC was utilized in ACC forming solution to maintain the same CaCO_3_: IC levels as those used in the other three forms of calcium carbonate. However, aragonite showed quite different kinetic adsorption curve compared to the others. The kinetic adsorption curve of IC onto aragonite (Figure 3b) reveals a continuous increase in q_t_ with time. After 60 min, the q_t_ of aragonite exceeds that of the other three forms of calcium carbonate, reaching more than 15 mg/g. Aragonite exhibited the highest q_t,_ among all forms of calcium carbonate. However, as shown in Table 1, aragonite exhibited a medium zeta potential, biggest diameter, and the lowest specific surface area among the four kinds of calcium carbonate, which cannot support the superiority of aragonite in adsorption of IC. As reported in literature, temperature also significantly influences adsorption process [26,27]. High temperature is a necessary condition for preparing aragonite (80 °C in this study), while all the other forms of calcium carbonate were prepared at ambient temperature. A previous study from our team has confirmed that the adsorption between MPs and calcium carbonate (mostly calcite) are endothermic process [13]. Since both IC and MPs are negatively charged in the co-precipitation solution, there is a high possibility that the adsorption between IC and aragonite is also endothermic reaction, which could explain the high q_t_ of aragonite in the kinetic experiment.

The pseudo-second-order model was applied to fit the four curves (Figure 3e), yielding R^2^ values greater than 0.9 for calcite, vaterite, and ACC, indicating good fitting. This suggests electrostatic attraction between calcium carbonate surface and IC, and also participation of chemisorption during the adsorption process [28,29]. However, for aragonite, the fitting was more consistent with pseudo-first-order model (Appendix A). This suggests that pore diffusion in the adsorbent is the decisive step in the whole adsorption process, rather than chemisorption domination.

A comparative assessment of surface adsorption and co-precipitation techniques was performed on crystalline calcium carbonates, focusing on their kinetic profiles as depicted in Appendix A. Owing to its pronounced instability in aqueous environments, amorphous calcium carbonate (ACC) was excluded from this analysis. As depicted in Appendix A, for each form of calcium carbonate, the q_t_ during the surface adsorption process is consistently lower than that observed in the co-precipitation process throughout the experiment, with the exception of the initial phase. The findings indicate that co-precipitation is markedly more effective than surface adsorption for immobilizing IC molecules. While both methods facilitate electrostatic attraction and chemisorption, co-precipitation additionally may employ a distinct mechanism—possibly a physical entanglement effect—to secure the IC molecules.

### 3.3. Isothermal Adsorption Analysis

The analysis of the kinetic adsorption curve revealed that crystalline calcium carbonate exhibited stronger adsorption compared to ACC, contrary to previous studies [30]. This discrepancy is attributed to the low IC concentration in ACC kinetic adsorption analysis. Thus, isothermal adsorption was conducted to reflect the maximum adsorption capacity of calcium carbonates to IC. Figure 4a–d shows the isothermal adsorption curves of IC onto the four forms of calcium carbonates. The curve of each calcium carbonate shows gradual-increase stage, plateau stage, and steep-increase stage successively. The gradual-increase stage represents the adsorption process, the plateau stage signifies the saturation of adsorption sites, and the steep-increase stage results from IC saturation in reaction solution. A study on IC adsorption onto aluminum hydroxide reported a similarly shaped isothermal adsorption curve [6]. Based on the plateaus of curves in Figure 4, the maximum amount of IC absorbed (q_e_ at plateau) is ~1100 mg/g for calcite, ~500 mg/g for aragonite, 3500 mg/g for the vaterite, and 6000 mg/g for ACC. ACC exhibited the highest q_e_ at plateau stage, confirming its superiority in adsorption capacity, which is likely attributed to its significantly higher surface area (Table 1).

The isothermal data was fitted into the Freundlich equation (Figure 4e) and Langmuir equation (Figure 4f). The results indicate that the adsorption of IC by the four forms of calcium carbonate is more consistent with the Freundlich model, suggesting multimolecular layer adsorption [31]. However, aragonite and vaterite exhibited lower R^2^ values, indicating that the adsorption of the two crystal forms is not a typical multimolecular layer adsorption.

### 3.4. FTIR Analysis

In Figure 5, the peaks of calcium carbonate near 710 cm^−1^, 1080 cm^−1^, and 1450 cm^−1^ are the C-O in-plane deformation vibrational peaks, the C-O stretching vibrational peaks, and the C-O antisymmetric stretching vibrational peaks, respectively [20]. Additionally, the peaks near 1797 cm^−1^ represent the stretching vibration of C=O, while the peaks near 860 cm^−1^ correspond to the CO_3_^2−^ out-of-plane deformation vibrational peaks [32]. The appearance of peaks in the spectra of the colorant lake near 1640 cm^−1^ (except for aragonite lake) represents a C=C stretching vibration, which is not present in spectra of calcium carbonate, indicating the successful adsorption of IC onto calcium carbonate particles. Appendix A shows further details of peak assignments. The spectra of different forms of calcium carbonate exhibit slight shifts and intensity changes in characteristic peaks attributed to differences in their lattice structures. Aragonite lakes did not show characteristic peaks of IC, consistent with previous XRD results likely due to minimal IC adsorption by aragonite. Isothermal adsorption results further confirm the relatively low adsorption efficiency of aragonite. Despite negligible IC structural changes in various crystalline calcium carbonates upon IC addition, there are discernible shifts in the characteristic peaks of IC in colorant lakes. Of particular significance is the observed shift in the spectral peak of IC from 1029 cm^−1^ to 1034 cm^−1^ within the colorant lakes, indicative of the characteristic vibrational frequency associated with the sulfonic acid group [17]. This peak shift corroborates the indication of chemisorption in kinetic analysis. The prominent peaks ranging from ~1418 to ~1495 cm^−1^ in the spectra of the four calcium carbonate samples correspond to the C-O antisymmetric stretching vibrations. Following the formation of the lakes, these peaks exhibit varying degrees of shifts, as evidenced by the altered curves in the lake spectra.

### 3.5. Raman Analysis

In Raman analysis, calcium carbonate crystals display characteristic peaks around 1100 cm^−1^ for symmetric stretching vibration (υ_1_) and around 700 cm^−1^ for double simplicity plane internal bending mode (υ_4_). The Raman spectra of calcium carbonate could be segmented into two sections, namely lattice modes below 400 cm^−1^ (Appendix A) and internal modes above 400 cm^−1^ (Appendix A) [33]. ACC displays typical amorphous characteristics with broad humps in the lattice mode of the Raman spectrum, consistent with typical characteristics of amorphous solids. The obtained spectra shapes for the three crystalline forms closely match those reported by Wehrmeister et al. [34]. These Raman spectral results help to confirm the predominant form of calcium carbonate in each sample, which is consistent with the conclusions based on XRD analysis.

In the internal mode (Appendix A), all crystalline calcium carbonates exhibit peaks at the spectral band of υ_1_ and a faintly visible spectral band of υ_4_, while only the band of υ_1_ is weakly visible in the spectrum of ACC. IC displays several distinct characteristic peaks near 1346 and 1576 cm^−1^. Despite the absence of IC detection in XRD and FTIR analyses for aragonite lakes, the appearance of IC’s characteristic peaks in the Raman spectra confirms pigment adsorption by aragonite, demonstrating higher sensitivity of Raman spectroscopy. Perhaps the primary reason for the superior sensitivity of Raman spectroscopy over the other two techniques is its capability to precisely target the sample’s surface using a microscope, with a focal point diameter that can be as small as 1 μm. Additionally, Raman spectroscopy is impervious to the intense absorption of water, thereby significantly preserving the sample’s original state [35,36,37]. With increase of IC concentration, the characteristic peaks of calcium carbonate in the Raman spectra of the colorant lakes weaken while the characteristic peaks of IC at 1346 and 1576 cm^−1^ strengthen and shift slightly in the direction of higher wavenumbers, indicative of sulfonic acid group participation in chemisorption [17]. Moreover, as more pigment is added to the colorant lake, a smaller shift in the pigment peak is observed. These results suggest that chemisorption involving sulfonic acid groups is relatively stronger at lower pigment content levels.

### 3.6. TGA-DSC Analysis

Figure 6 shows the TGA-DSC curves of the four calcium carbonates and their corresponding colorant lakes. The thermogravimetric curve illustrates the sample’s weight loss as temperature increases. Figure 6 illustrates that all calcium carbonate samples undergo two distinct phases of weight loss at varying rates. The first phase, predominantly due to water evaporation, continues up to around 600 °C, while the second phase, from around 600 °C to 800 °C, is attributed to the decomposition of calcium carbonate. Specifically, the first phase curves of ACC, aragonite and vaterite can be segmented further into a rapid-weight-loss segment due to free water evaporation, followed by a gradual-weight-loss segment as bound water dissociates from the samples. In the TGA curve of ACC (Figure 6a), the rapid-weight-loss segment extends to approximately 200 °C, aligning closely with results reported by Schmidt et al. [38]. The rapid-weight-loss segment of aragonite’s curve (Figure 6e) reaches about 350 °C, and that of vaterite’s (Figure 6g) reaches about 250 °C. However, calcite (Figure 6c) exhibits a nearly uniform rate of weight loss during their first phases of weight loss. TGA curves for the four types of calcium carbonate are extensively documented in the literature [38,39,40,41]. While these curves are typically similar in the second stage, they exhibit notable diversity in the first stage, presenting a range of curve shapes specific to each type. The observed variations in the first stage of the TGA curves can be attributed to the disparate preparation methods employed for calcium carbonate across various laboratories. These methods may alter the micro-morphology and hydration state of the calcium carbonate, which are critical factors deciding the first stage of the curve, yet they do not affect the inherent chemical composition of calcium carbonate that determines the second stage of the curve. For all the calcium carbonates, the incorporation of IC resulted in a more pronounced weight loss before 600 °C, as indicated by the TGA curves of the lakes, particularly for those with the highest IC addition (500 mg, 250 mg or 33.3 mg). This observation may be attributed to the additional water introduced by IC or the thermal decomposition of IC, which typically commences at temperatures above 350 °C [42].

In the DSC curves, two characteristic peaks are observed across all calcium carbonate samples and their corresponding lakes, occurring at approximately 50 °C and 700 °C. These peaks are indicative of the rapid evaporation of free water and the decomposition of calcium carbonate, respectively. Within the literature, there is a reported variation in the specific temperatures corresponding to the two characteristic peaks of calcium carbonate. This variability can be attributed to the diverse methodologies and parameters utilized in the synthesis of calcium carbonate. [38,39]. Notably, the sharp exothermic peak at around 350 °C in the DSC curve of ACC (Figure 6b) signifies the crystallization of ACC into calcite [38]. Similarly, the exothermic peak at about 425 °C in the DSC curve of vaterite (Figure 6h) marks the phase transition from vaterite to calcite [43]. The higher temperature of vaterite’s exothermic peak (425 °C) compared to ACC’s (350 °C) corroborates vaterite’s greater thermal stability. No crystallization peak is present in the DSC curves of aragonite (Figure 6f) and calcite (Figure 6d), denoting their superior thermal stability relative to ACC and vaterite. The presence of a single exothermic peak implies homogeneity in the ACC particles [36]. Comparing to curves of the pure calcium carbonates, an additional endothermic peak is observed at approximately 550 °C in curves of ACC lake, calcite lake and vaterite lake, potentially due to the thermal decomposition of IC. It is noteworthy that only the aragonite lakes lacked the exothermic peak at approximately 550 °C, which could be attributed to the lower IC concentration in aragonite lakes, arising from aragonite’s reduced adsorption capacity.

### 3.7. EDS Analysis

To elucidate the distribution of IC within the colorant lakes, the sulfur element—exclusive to the pigment—was analyzed using EDS on both the surface and interior (via cross-sections) of the lake particles. Owing to the diminutive size and dense texture of ACC, cross-sections were unattainable, precluding ACC from this phase of EDS analysis.

The elemental distribution on both the surfaces and cross-sections of the three crystalline forms of calcium carbonates and their corresponding lakes is depicted in Appendix A, with representative examples of surfaces and cross-sections illustrated in Appendix A. Consistent with expectations, sulfur, a marker for the presence of IC, was not detected on the surfaces or cross-sections of the pure calcium carbonate samples. Notable sulfur content was observed on the surfaces of calcite lake and vaterite lake (1.80% and 1.96%, respectively), whereas aragonite surfaces exhibited no sulfur, likely reflecting aragonite’s lower IC adsorption capacity. Cross-sectional analysis revealed sulfur solely in calcite lake (0.02%), suggesting the presence of IC both on the surface and within the interior of calcite lake particles, predominantly on their surface. The challenge of detecting sulfur via EDS on cross-sections stems from the difficulty in preparing suitable cross-sections and the serendipitous nature of locating such sections under SEM. In this study, out of five suitable calcite lake particle cross-sections identified, sulfur was found in only two. This suggests an uneven distribution of IC within the particles, or the presence of the pigment solely in a subset of the particles. Either scenario could result in the inability of EDS to detect the pigment. Hence, the absence of sulfur on the cross-sections of the other two calcium carbonate lake particles does not preclude the potential internal presence of the pigment. Anyhow, the findings provide compelling evidence for the existence of IC on the surface and in interior of calcite lake particles.

## 4. Conclusions

In this study, we synthesized various forms of calcium carbonate and effectively integrated them with IC using a co-precipitation approach to create colorant lakes. The addition of IC generally maintained the crystalline structure of calcium carbonate, with the exception of aragonite, which transitioned to exhibit a substantial vaterite presence. While the three crystalline forms of calcium carbonate produced micrometer-sized particles, ACC formed nanoparticles, resulting in the highest specific surface area. Kinetic analysis revealed that aragonite adhered to a pseudo-first-order model, whereas the other forms aligned more with a pseudo-second-order model, indicative of chemisorption. These findings were supported by FTIR and Raman spectroscopy, which verified the involvement of sulfonic acid groups in chemisorption. A comparative study of surface adsorption and co-precipitation methods showed that co-precipitation was superior for IC adsorption. Isothermal adsorption analysis demonstrated that ACC had the greatest IC loading capacity, followed in order by vaterite, calcite, and aragonite. The adsorption behavior of all four calcium carbonates was more consistent with the Freundlich isotherm model over the Langmuir model, suggesting a multilayer adsorption process. The incorporation of IC markedly altered TGA and DSC profiles of the calcium carbonates by introducing additional water content and impacting their microstructure. EDS analysis revealed that IC was primarily located on the surface of the lake particles with a minor intraparticle presence. These findings underscore the necessity for further research into the efficacy of different crystalline forms in actual food applications.

## Figures and Tables

**Figure 1 foods-13-02607-f001:**
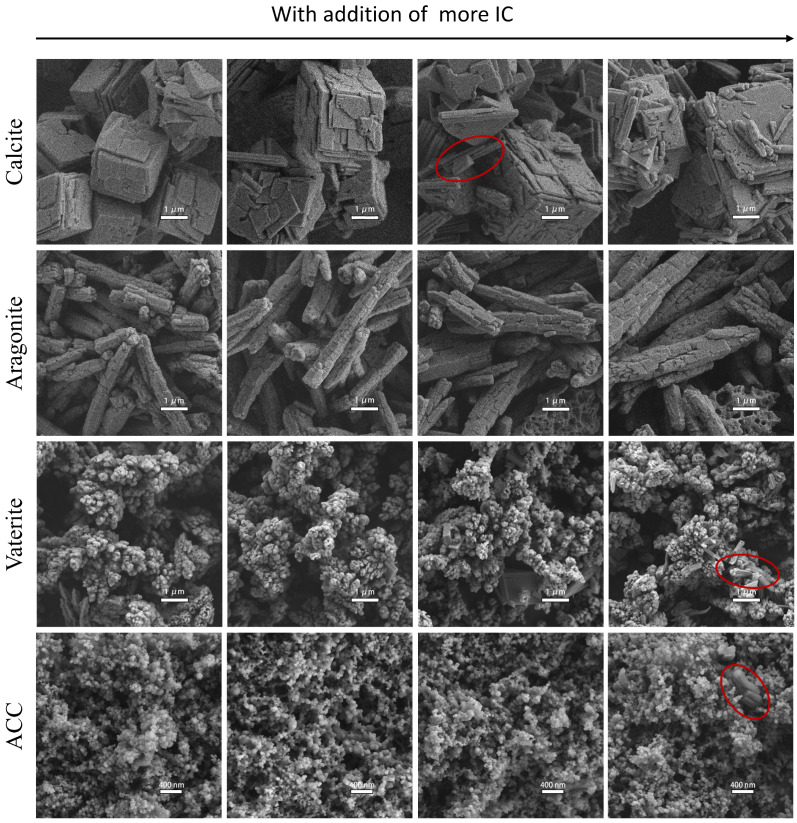
SEM images of four kinds of calcium carbonates and their colorant lakes prepared with addition of different amount of IC. The red circles circle the bars.

**Figure 2 foods-13-02607-f002:**
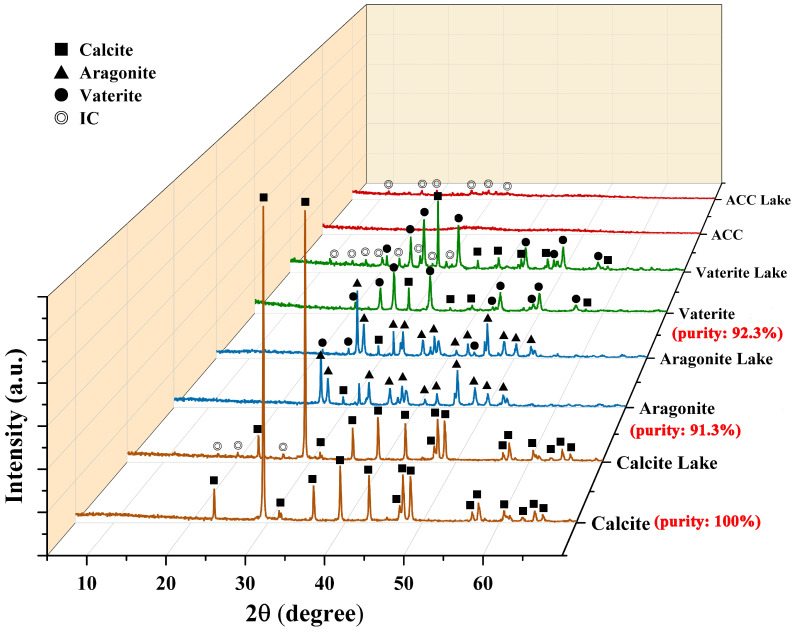
XRD diffractogram of different calcium carbonates and their colorant lakes (■: Calcite, ▲: Aragonite, ●: Vaterite, ◎: IC, the red font indicates the purity of the crystal form).

**Figure 3 foods-13-02607-f003:**
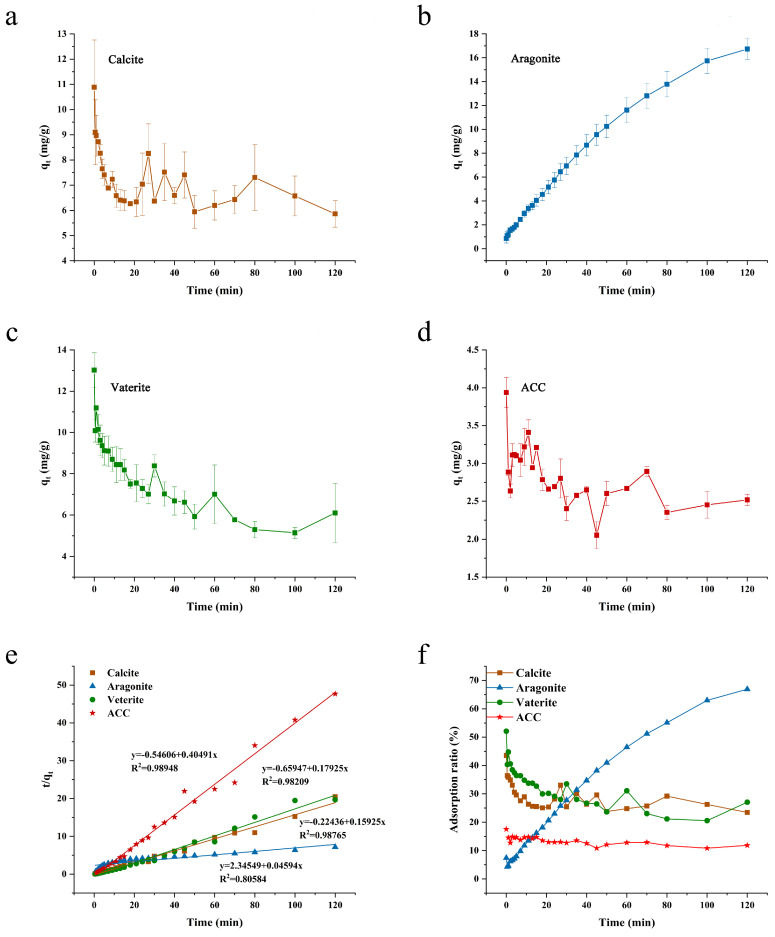
Kinetic curves for adsorption of IC by the four calcium carbonates using co-precipitation method ((**a**): Calcite; (**b**): Aragonite; (**c**): Vaterite; (**d**): ACC), fitting of the adsorption data into pseudo -second order model (**e**) and curves of adsorption ratio change with time (**f**).

**Figure 4 foods-13-02607-f004:**
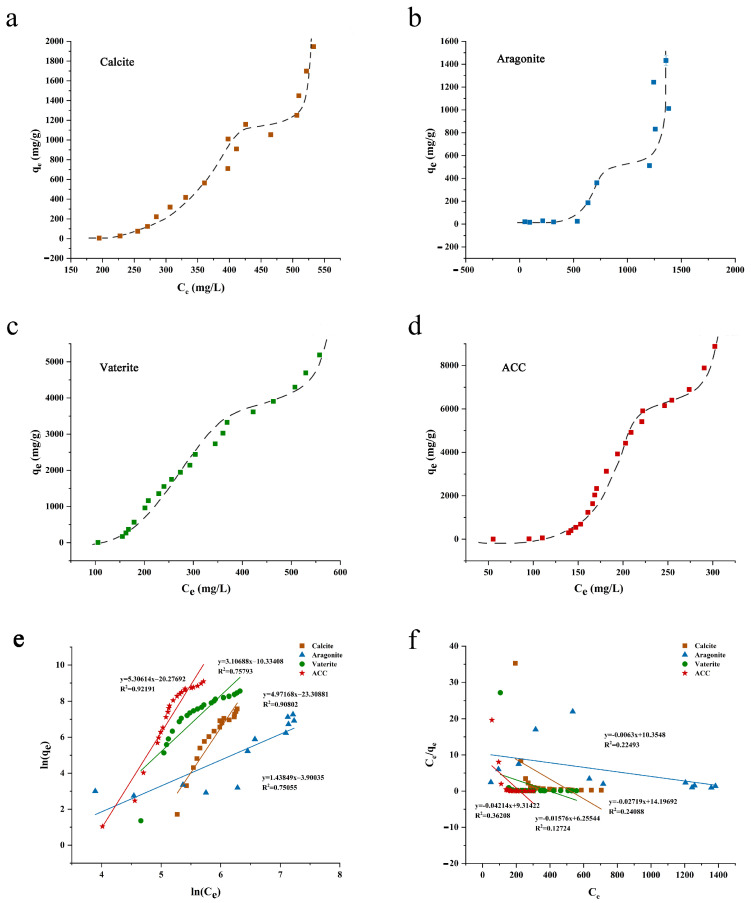
Isothermal adsorption curves ((**a**): Calcite; (**b**): Aragonite; (**c**): Vaterite; (**d**): ACC), fitting of isothermal data into Freundlich equation (**e**) and fitting of isothermal data into the Langmuir equation (**f**) (The hatched lines are visual fit only).

**Figure 5 foods-13-02607-f005:**
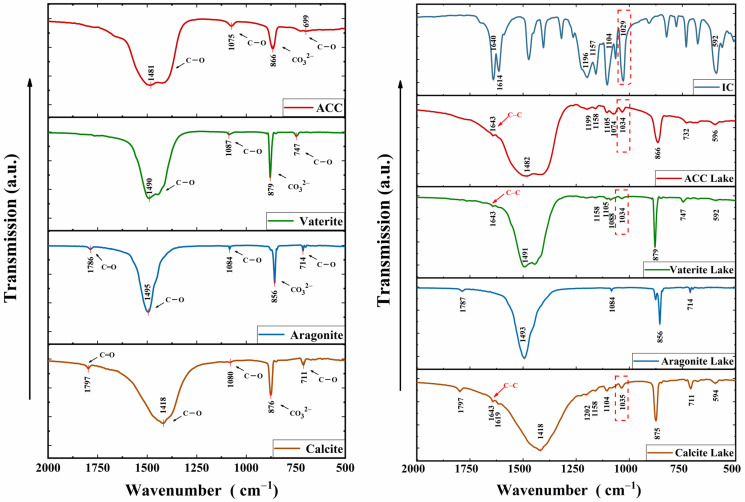
Infrared spectra of IC, different forms of calcium carbonates and their colorant lakes (Red dashed box shows typical peak shift).

**Figure 6 foods-13-02607-f006:**
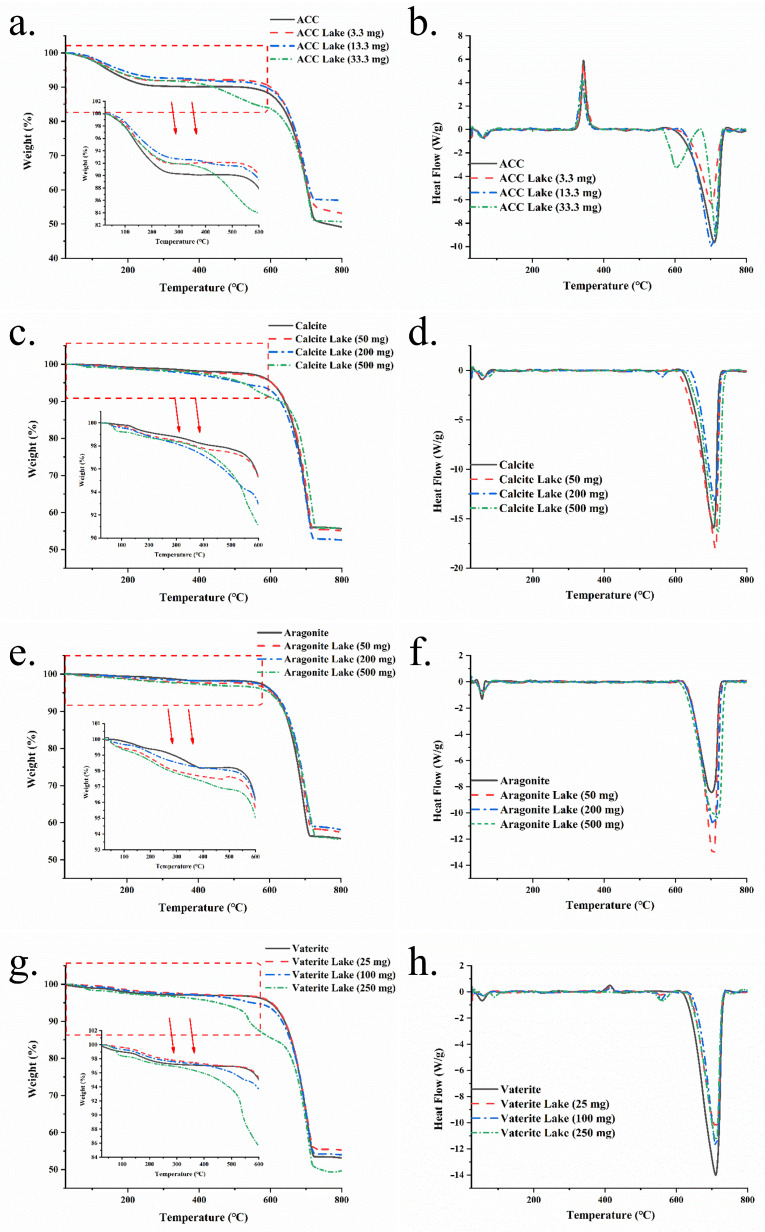
TGA curves (**a**,**c**,**e**,**g**) and DSC curves (**b**,**d**,**f**,**h**) of the four forms of calcium carbonates and their colorant lakes.

**Table 1 foods-13-02607-t001:** Zeta potential, average diameter, specific surface area and ∆b* of different calcium carbonates and their colorant lakes.

Samples	Zeta Potential (mV)	Average Diameter (μm)	Specific Surface Area (m^2^/g)	Δb*
Calcite	−21.57 ± 0.76 ^h^	6.116 ± 0.27 ^c^	3.985 ± 0.035 ^j^	-
Calcite Lake (50 mg)	−21.33 ± 1.04 ^h^	5.794 ± 0.10 ^d^	3.829 ± 0.159 ^j^	0.01 ± 0.095 ^c^
Calcite Lake (200 mg)	−15.27 ± 0.87 ^f^	6.459 ± 0.01 ^b^	2.905 ± 0.145 ^k^	0.07 ± 0.026 ^b^
Calcite Lake (500 mg)	−10.16 ± 0.73 ^e^	4.549 ± 0.13 ^f^	3.831 ± 0.081 ^j^	0.07 ± 0.045 ^b^
Aragonite	−5.867 ± 0.67 ^c^	7.564 ± 0.06 ^a^	3.841 ± 0.021 ^j^	-
Aragonite Lake (50 mg)	2.25 ± 0.30 ^a^	4.486 ± 2.31 ^f^	5.079 ± 0.119 ^h^	0.04 ± 0.005 ^c^
Aragonite Lake (200 mg)	3.53 ± 0.60 ^a^	4.080 ± 3.93 ^g^	4.449 ± 0.029 ^i^	−0.03 ± 0.015 ^c^
Aragonite Lake (500 mg)	−0.621 ± 0.91 ^b^	4.831 ± 0.05 ^e^	5.086 ± 0.074 ^h^	−0.10 ± 0.030 ^c^
Vaterite	−8.79 ± 0.54 ^d^	3.188 ± 0.10 ^h^	7.432 ± 0.082 ^g^	-
Vaterite Lake (25 mg)	−10.05 ± 0.46 ^e^	6.655 ± 0.03 ^b^	8.036 ± 0.024 ^f^	0.08 ± 0.003 ^b^
Vaterite Lake (100 mg)	−9.51 ± 0.25 ^e^	4.272 ± 0.02 ^f^	8.182 ± 0.032 ^f^	−0.08 ± 0.015 ^d^
Vaterite Lake (250 mg)	−19.07 ± 0.78 ^g^	2.324 ± 0.01 ^j^	12.042 ± 0.052 ^e^	−0.14 ± 0.010 ^e^
ACC	−1.00 ± 0.72 ^b^	0.368 ± 0.02 ^k^	36.038 ± 0.058 ^b^	-
ACC Lake (3.3 mg)	2.74 ± 0.54 ^a^	0.175 ± 0.01 ^l^	38.375 ± 0.045 ^a^	−0.06 ± 0.105 ^d^
ACC Lake (13.3 mg)	2.96 ± 0.81 ^a^	0.524 ± 0.02 ^k^	32.029 ± 0.068 ^c^	−0.04 ± 0.020 ^d^
ACC Lake (33.3 mg)	0.34 ± 1.21 ^b^	0.391 ± 0.01 ^k^	29.249 ± 0.259 ^d^	−0.10 ± 0.030 ^d^
IC	-	-	-	0.14 ± 0.020 ^a^
Calcite + IC *	-	-	-	−0.08 ± 0.133 ^d^
Aragonite + IC *	-	-	-	0.07 ± 0.023 ^b^
Vaterite + IC *	-	-	-	−0.17 ± 0.035 ^e^
ACC + IC *	-	-	-	0.17 ± 0.017 ^a^

The values with different superscript letters in the same column are significantly different (*p* < 0.05). * The samples were prepared by direct mixing of calcium carbonate and IC (CaCO_3_:IC = 3 g:500 mg).

## Data Availability

The original contributions presented in the study are included in the article and Appendix A, further inquiries can be directed to the corresponding author.

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
