# Peer review of "Influence of the Crystal Forms of Calcium Carbonate on the Preparation and Characteristics of Indigo Carmine-Calcium Carbonate Lake"

_foods, 2024, doi:10.3390/foods13162607_

Round 1

Reviewer 1 Report

Comments and Suggestions for Authors

The manuscript is professionally structured and organized to present the study's results. Substantial experimental work is performed, although some additional explanations and corrections would be necessary.

1.     It would be good to provide a SEM image of the Indigo Carmine (IC).

2.     XRD: the term “spectrum” is not applicable here. The term "spectrum" refers to frequency, wavelength, and energy-dependent plots/presentations. In the case of X-ray diffraction, one can also say "diffraction pattern", but the term "diffractogram" is also correct.

3.     XRD of IC crystal solid would contribute to easier follow-up explanation.

4.     Table 1: Unfortunately, I do not see Table 1, which is necessary to follow a large part of the Manuscript.

5.     When considering the resolution of the FTIR spectrum, it is better to give the wavenumbers without decimal places.

6.     Figures 5 and 6 should be present as one Figure to avoid repeating the spectra and especially to make it easier to follow the changes in the spectra. It is sufficient to show the spectra in the spectral interval below 2000 cm-1.

7.     What is the size of the surface of the sample analyzed by Raman spectroscopy in one spectrum? Was a microscope used in the selection of the surface? Perhaps this is the reason for the "higher sensitivity" of Raman spectroscopy compared to bulk methods FTIR and XRD. With spectra, it is very vague/impractical to use the expression "shift slightly to the right". The terms “Bathochromic shift or a red shift and a hypochromic shift or a blue shift, or simply shifted to lower or higher wavenumbers” are much more specific, because very often (and probably more regularly) Raman and FTIR spectra are displayed from higher to lower wavenumbers.

8.     The Raman spectra in Fig. S4 should be presented so that the changes mentioned in the text are clearly visible to avoid speculation. On some spectra, the position of the band seems not precisely determined. It is also sufficient to give wavenumbers without decimal places.

9.     Figure 7. The letters a-h are not mentioned either in the text or in the Figure title. What is their purpose?

10.  Replace ml with mL, please.

I believe that after a major revision, the paper can be considered in this highly respected journal.

Author Response

Comment 1. It would be good to provide a SEM image of the Indigo Carmine (IC).

Respond: Thanks for the valuable feedback on our manuscript. A SEM image of the IC has been included in the supplementary file (Figure S1a) and is described in the revised manuscript (line 224-226).

Comment 2. XRD: the term spectrum is not applicable here. The term "spectrum" refers to frequency, wavelength, and energy-dependent plots/presentations. In the case of X-ray diffraction, one can also say "diffraction pattern", but the term "diffractogram" is also correct.

Respond: Thanks for the suggestion. As suggested, "spectrum" has been replaced by "diffractogram" (line 231-254).

Comment 3. XRD of IC crystal solid would contribute to easier follow-up explanation.

Respond: Thanks for the valuable comment. XRD diffractograms of IC have been added to the supplementary file (Figure S1b) and is described in the revised manuscript (line 231-232).

Comment 4. Table 1: Unfortunately, I do not see Table 1, which is necessary to follow a large part of the Manuscript.

Respond: Thanks for the reminder. Table 1 has been added to the manuscript (line 283-286).

Comment 5. When considering the resolution of the FTIR spectrum, it is better to give the wavenumbers without decimal places.

Respond: Thanks for the suggestion, and wavenumbers without decimal places have been added in Figure 5, Table S2, Figure S4 and Figure S5.

Comment 6. Figures 5 and 6 should be present as one Figure to avoid repeating the spectra and especially to make it easier to follow the changes in the spectra. It is sufficient to show the spectra in the spectral interval below 2000 cm-1.

Respond: Thanks for the suggestion. The figure 5 and 6 have been combined to create a new Figures 5 showing spectra in the spectral interval below 2000 cm-1.

Comment 7. What is the size of the surface of the sample analyzed by Raman spectroscopy in one spectrum? Was a microscope used in the selection of the surface? Perhaps this is the reason for the "higher sensitivity" of Raman spectroscopy compared to bulk methods FTIR and XRD. With spectra, it is very vague/impractical to use the expression "shift slightly to the right". The terms “Bathochromic shift or a red shift and a hypochromic shift or a blue shift, or simply shifted to lower or higher wavenumbers” are much more specific, because very often (and probably more regularly) Raman and FTIR spectra are displayed from higher to lower wavenumbers.

Respond: Thanks for your insightful comments, and the responds are as follows:

(1) An explanation of the reasons for the higher sensitivity of Raman spectroscopy has been added to the manuscript (Line 445-449).

(2) The expression "shift slightly to the right" has been changed to " shift slightly in the direction of higher wavenumbers" (line 451-452).

(3) Thanks for the suggestion, and all the Raman and FTIR spectra (Figure 5, Figure S4-5) in this manuscript have been modified with X-axis from higher to lower wavenumbers.

Comment 8. The Raman spectra in Fig. S4 should be presented so that the changes mentioned in the text are clearly visible to avoid speculation. On some spectra, the position of the band seems not precisely determined. It is also sufficient to give wavenumbers without decimal places.

Respond: Thanks for the valuable comments. If we have correctly understood the comments, our responds are as follows:

The positions of υ1 and υ4 are marked in Figure S4 (Named Figure S5 in the revised manuscript) and two points are indicated by red dashed boxes: a. The appearance of the peaks of IC in the aragonite lake; b. The slight shift of the characteristic peaks of IC at 1346 and 1576 cm-1 towards the higher wavenumbers. In addition, the wavenumbers are now given without decimal places.

Comment 9. Figure 7. The letters a-h are not mentioned either in the text or in the Figure title. What is their purpose?

Respond: Thanks for the reminder. The names of Figure 7a to Figure 7h (Named Figures 6a-h in the revised manuscript) have been added into the text where the corresponding data is mentioned or discussed (line 465-499). Besides, the introduction on a-h figures has also been added to the figure title (line 484).

Comment 10. Replace ml with mL, please.

Respond: Thanks for the valuable comments. All the “ml” in this manuscript have been replaced with “mL”.

Reviewer 2 Report

Comments and Suggestions for Authors

The manuscript foods-3138246 constitutes a rigorous and very appropriate investigation for the FOODS magazine, which stands out for its approach, the richness of the experimental part and the very appropriate use of spectral techniques and thermal stability studies of the calcium carbonate samples in him included.

Minor points:

1. I wonder if bibliographical citations should not appear as is accepted in almost all scientific journals, i.e. [1] for the first reference,...

2. 'The BET test' will be related to the multi‐point Brunauer‐Emmett‐Teller specific surface area testing methodology.

3. Insert a blank space between the numerical temperature value and the unit symbol (C). Currently this white space is only omitted white between a numeric value and the % symbol.

Author Response

Comment 1. I wonder if bibliographical citations should not appear as is accepted in almost all scientific journals, i.e. [1] for the first reference, ...

Respond: Thanks for this reminder. As suggested, citation style with numbers has been employed in this manuscript.

Comment 2. 'The BET test' will be related to the multi‐point Brunauer‐Emmett‐Teller specific surface area testing methodology.

Respond: Thanks for the comment. The “BET” has been clearly specified in the manuscript (line 118).

Comment 3. Insert a blank space between the numerical temperature value and the unit symbol (⁰C). Currently this white space is only omitted white between a numeric value and the % symbol.

Respond: Thanks for the reminder. As suggested, a blank space has been inserted between the numerical temperature value and the unit symbol (⁰C) all through the manuscript.
